# A Biomechanical Analysis of the Influence of the Morfology of the Bone Blocks Grafts on the Transfer of Tension or Load to the Soft Tissue by Means of the Finite Elements Method

**DOI:** 10.3390/ma15249039

**Published:** 2022-12-17

**Authors:** Blanca Gil-Marques, Antonio Pallarés-Sabater, Aritza Brizuela-Velasco, Fernando Sánchez Lasheras, Pedro Lázaro-Calvo, María Dolores Gómez-Adrián, Carolina Larrazábal-Morón

**Affiliations:** 1Doctoral School, Catholic University of Valencia San Vicente Mártir, 46001 Valencia, Spain; 2School of Dentistry, Departament Medical and Surgery, Catholic University of Valencia San Vicente Mártir, 46001 Valencia, Spain; 3Departament Endodonthics, School of Dentistry, Catholic University of Valencia San Vicente Mártir, 46001 Valencia, Spain; 4Departament of Dentistry, Universidad Europea Miguel de Cervantes, 47012 Valladolid, Spain; 5Department of Mathematics, Faculty of Sciences, University of Oviedo, 33007 Oviedo, Spain; 6Institute of Space Sciences and Technologies of Asturias (ICTEA), University of Oviedo, 33004 Oviedo, Spain; 7Departament of Periodonticx, Complutense University, 28004 Madrid, Spain

**Keywords:** finite element analysis, oral bone block, split bone block, regenerative oral surgery, primary wound closure, wound dehiscence

## Abstract

Edentulism produces resorption of alveolar bone processes, which can complicate placement of dental implants. Guided bone regeneration techniques aim to recover the volume of bone. These treatments are susceptible to the surgical technique employed, the design of the autologous block or the tension of the suture. These factors can relate to major complications as the lack of primary closure and dehiscence. The present study, using finite element analysis, aimed to determine differences in terms of displacement of the oral mucosa, transferred stress according to Von Mises and deformation of soft tissue when two block graft designs (right-angled and rounded) and two levels of suture tension (0.05 and 0.2 N) were combined. The results showed that all the variables analyzed were greater with 0.2 N. Regarding the design of the block, no difference was found in the transferred stress and deformation of the soft tissue. However, displacement was related to a tendency to dehiscence (25% greater in the right-angled/chamfer design). In conclusion different biomechanical behavior was observed in the block graft depending on the design and suture tension, so it is recommended to use low suture tension and rounded design. A novel finite element analysis model is presented for future investigations.

## 1. Introduction

Diet is the fundametal pillar of health in general and it has always been believed that good health starts with adequate nutrition and a correct swallowing of foodstuffs [1]. The loss of teeth provokes a natural reshaping of the alveolar ridge, which provides a challenge to the fitting of dental implants; the latter has become a highly popular treatment in everyday dentistry practice due to the success of long-term evaluations and its clinical, biological and mechanical advantages [2,3,4]. However, the aforementioned loss of bone volume makes it necessary for us to perform regenerative bone surgery in order to reconstruct the alveolar ridge and so be able to offer optimal long-term results with dental implants, thus providing patients with a higher quality of life by carrying out a fixed oral rehabilitation which allows them to eat, talk and bite normally [5,6].

To date many different bone regeneration techniques have been developed, with autologous bone or with bone substitutes (xenografts, allografts and alloplastic material) [7]. Autologous bone grafts are considered the “gold standard” owing to their properties of osteoconduction, osteoinduction and osteogenesis; however, one worry is the stability of the graft in the long term [7,8,9]. The most widely used techniques for guided bone regeneration are the expansion of the ridge and block grafts [10]. In 2015, Khoury & Hanser put forward a modification to the block autologous graft technique called Split Bone Block (SBB) which employs a combination of very thin blocks (1 mm thick) to form a casing around the implant. The SBB technique has a great advantage over conventional block techniques in that there is greater vascularization due to the thinness of the block, which prevents the resorption of bone particles and thus giving it stability over time [11].

One of the main reasons why bone regeneration fails is related to its exposure to the graft, which can cause it to become contaminated or produce hydrolisis, jeopardizing the success of the process [12,13]. To avoid this, primary wound closure is most important, manipulating the soft tissue properly in order to achieve a closure without stress during the healing period [14,15]. It is worth noting that the increase in volume caused by the graft itself in the receptor site tends to generate stress in the repositioned soft tissue during the healing period and therefore open the wound or cause dehiscence [16]. In this respect, De Stavola y Tunkel (2013) observed that stress inferior to 5 g on the repositioned gum did not cause a tendency to dehiscence of the surgery wound. For their part, Burkardt & Lang (2009) [16] observed that stress from 0.01–0.15 N generated a lower rate of dehiscence and that in turn, stress greater than 0.1 N increased the percentage of dehiscence significantly. In recent years various articles have been published on the manipulation of flaps to prevent dehiscence and achieve primary wound closure, although their evidence is limited given that they refer only to standard case studies [12,17,18,19].

Jensen & Terheyden (2009) consider that another of the variables that could cause dehiscence is the morphology of the block. These authors led a clinical study whose results showed, on the one hand, that regenerations of large areas present a greater contact between the gum and the regeneration and so there is a greater prevalence of dehiscence, and also that a regeneration morphology which protrudes from the bone framework can produce a greater displacement of the oral mucosa. Finally, the results showed that vertical regenerations showed a greater tendency to dehiscence owing to the fact that a larger area of gum needs to be in contact with the regeneration; however, no differences were found regarding volume gained, stability of the graft or success of the implant whether an absorbable or non-absorbable membrane is used to cover the defective area [7].

In short, the suture tension applied and the morphology of the graft can be two independent variables which affect the biomechanical behaviour of the surgical wound during the SBB regeneration process, especially in the first few hours of the process, which coincide with the inflammatory phase. In turn, this biomechanical behaviour may be closely related to a tendency to dehiscence and a failure to achieve primary wound closure. Nonetheless, these results can prove difficult to verify in clinical models or in vivo, where other co-variables or confounding variables may also be present. This circumstance is common to other biomechanical suppositions in the field of dentistry and more specifically, in that of oral implantology. In this sense the analysis by means of the finite elements method (FEM) is currently the most widely-used method in science and engineering with the aim of simulating the behaviour of systems subjected to loads and deformations. More specifically, with the aid of FEM it has been possible to resolve physical problems and, in the context which concerns us here, to determine the biomechanical behaviour of systems of interest [20]. In the field of oral surgery it has been extensively applied in the study of the biomechanical behaviour of implant-supported prosthetics in the jaw bone, as well as in the distribution of transferred stress on the supporting bone and adjacent structures [21]; despite this, we have found no study which analyses the behaviour of soft tissue during its healing phase, let alone when the jaw bones are subject to bone regeneration. In this regard, after going through existing scientific literature we have found no evidence of the biomechanical behaviour shown by the gum under the stress of regeneration with cortical bone blocks. For this reason, the justification of this research is to clarify this question by means of a finite element study.

The aim of this study was to establish and compare the stress, deformation and displacement transferred to soft tissue when two levels of suture tension are applied (0.05 N, 0.2 N) to two different graft designs (right-angled edge and rounded edge), in a GBR procedure using an autologous bone graft block.

## 2. Materials and Methods

This section describes the FEM models employed in the problem under study. As it is well-known, A FEM model requires a defined geometry that in the case of the present research is an atrophied ridge of the upper jawbone, on which the regeneration of the defective bone is performed, a definition of the parameters characterizing the study model, some boundary conditions that mimic how the jaw bone behaves and, also, the definition of some loads that match with the biomechanics of the study.

### 2.1. Model Geometry

The three-dimensional (3D) geometry of this research simulates the bone in the premolar-molar region, with an area of horizontal bone defect, in which the most coronal part of the ridge was 3 mm wide in the buccal-palatal direction and the most apical part 5 mm wide in the buccal-palatal direction. The section of bone to be re-modelled was created by using a CBCT (Cone Beam Computed Tomography) from an edentulous patient in the premolar area with sections of less than 0.25 mm [22], from which a DICOM (Digital Imaging and Communications in Medicine) was exported; from this a section of jaw bone was created with a density of type A2 according to Lekholm & Zarb’s classification (1985) [23], which existing literature shows to be the most common in the jaw [24], with a 1–2 mm layer of compact bone surrounding a core of trabecular bone [22,25].

The autologous bone block was also modelled according to Khoury’s SBB technique [11]. Following this technique, two blocks of cortical bone 1 mm thick were created, the first of which being placed in the buccal area of the defect, and the second in its crestal area thus creating a casing. Two suppositions were simulated, one with blocks with rounded edges (R), (Figure 1) and the second with blocks with right-angled edges/Chamfer (C) (Figure 2).

These thin bone blocks were fixed into the jaw by means of two microscrews, 1.2 mm in diameter and 10 mm long, (Stoma por Prof. Khoury), made from a surgical steel alloy, hardened to prevent them breaking or bending [10,11].

The gap created between the blocks and the jaw was filled with autologous particles (trabecular bone) of type A4 [26]. The model employed uses the hypothesis that an increase in suture tension does not mean greater stress and deformation transferred to soft tissue or a greater displacement thereof in a GBR procedure using an autologous bone graft block.

Lastly, the oral mucosa covering the block graft was modelled with a thickness of 1.5 mm [25,27], taking into account that the chewing mucosa (attached gingival) showed visco-elastic behaviour and resistance to deformation under the applied load, thanks to being firmly fixed to the rigid cortical bone.

### 2.2. Properties of Materials

The properties of the materials employed in the FEM models were taken from existing literature [27,28,29,30,31] and are listed in Table 1.

On the basis of the considerations of previous studies [32,33,34] and with the aim of simplifying the model, the maxillary bone crest, microscrews, cortical layers and bone particles were assumed to be homogenous, isotropic and with linear elasticity. It was further considered that the mucosa was fixed to the cortical bone and was assumed to be homogenous, isotropic and viscoelastic.

Taking into account the scientific literature, it was assumed that the microscrews were totally osseointegrated into the cortical blocks of the graft and the remaining bone of the jaw, considering the osseointegration to be the direct, structural and functional connection between the living bone and the surface of a microscrew [35,36,37]. Those areas not affecting the surgery are defined as fixed nodes and zero displacement was assumed for them [25,38].

### 2.3. Loads and Boundary Conditions

3D modelling was then performed, making use of the two designs of the thin bone blocks (right-angled and rounded edges) [10] and two tensile forces applied to the stitches of the incision (Figure 1, Figure 2 and Figure 3) In this manner, four test models were obtained: a first model with the cortical blocks with rounded edges (R) and a tensile force on the suture of 0.05 N, a second model which presented the same morphology as the previous one but subjected to a force of 0.20 N, and the last two in which the previous morphology was modified, simulating two cortical blocks at a right angle/chamfer (C) with a tensile force on the suture of 0.05 N and 0.20 N, respectively.

In order to perform the finite element tests, the average force generated on the oral mucosa during the healing phase was applied. The system of forces in the incision-suture area is very complex, and we opted for a simplified representation of it. Static loads were used, starting with a tension of 5 g [19], as well as a tension of between 0.05 and 0.20 N to evaluate the stress on the wound [16]. This force was applied on the line of suture (red line) at an angle of 45° on the XY axis, and was carried out in tabular form depending on time [27] (Figure 4).

After performing the FEA, the results of von Mises stress criterion, maximum deformation and the gum displacement were analyzed.

### 2.4. Convergence Test

Finally, and in order to verify the FEM analysis, a convergence test was performed. For this purpose, the size of the mesh was analysed, making use of three different densities of mesh, with the aim of testing the lack of influence on said mesh. Just as the size of the average element is very important, the aspect ratio of the meshed element is also of great importance.

Three meshes (A–C) were created according to their density, with C having the greatest density. For the latter, a target size of 0.3 mm was imposed, with a maximum of 0.6 mm, a growth rate of 1.5 and a singularity removal algorithm of 1.5∙× 10^−3^ mm. In other words, any singularity of less than 1.5 microns was omitted so as to avoid convergence problems. On average, 125,000 nodes were created in each model to perform the FEM. (Ansys 2020 R1; Ansys Inc., Canonsburg, United States).

## 3. Results

### 3.1. Von Mises Stress Distribution on the Jaw Bone

The Von Mises stress distribution on the bone was analysed for the four suppositions of morphology of the blocks and applied stress, and the results can be seen in Table 2.

On analysing the results, we failed to observe any appreciable difference in terms of the pressure exerted on the oral mucosa by the blocks according to the different shape of the bone blocks (R and C); in both cases, practically identical results were obtained with the same force. However, the applied stress was analysed for each of the models, and here it was observed that in both models (R and C), with an applied stress of 0.05 N its Von Mises stress was significantly lower when compared to the same models with stress of 0.2 N applied to the suture. For all four suppositions, the greatest stress generated on the bone was found to be in the crestal area of the defect, between the microscrew, the adjacent molar and the defect itself.

Comparing the average values of the blocks, whether rounded or right-angled, at 0.2 N and 0.05 N, we observed that they showed a percentage difference of 75% more stress on the incision area when 0.20 N was applied (Figure 5).

### 3.2. Von Mises Stress Distribution in the Gum

Von Mises stress distribution in the oral mucosa was analysed for the four morphologies of the blocks and stress under analysis, as was also done in the Von Mises analysis of the bone. When the results were analysed, it was observed that there were similar results to those of the transferred stress to the jaw bone, therefore no appreciable differences were observed in terms of the shape (C and R); in both cases they gave nearly identical results with the same force. However, once the applied stress for both models had been analysed, it was observed that for each of the models (R and C), with an applied stress of 0.05 N its Von Mises stress is significantly lower when compared to the same models with a stress of 0.20 N applied to the suture (Table 2).

A 75% more of force was obtained in the 0.2 N models, comparing the value of generated stress between the stress model 0.2 N (R and C) and the value obtained for the 0.05 N stress models (R and C).

On analyzing the stress distribution map, it was observed that in all cases, the greatest values of the distribution of stress on the gum was to be found in the edentulous area, that is, in the crestal area above the defect, although on the stress distribution map a lesser distribution of stress could be observed for the assumed rounded angle with an applied stress of 0.05 N.

With regard to the results obtained, the behaviour of the gum could best be appreciated on the inside in the force relaxation area, or on the outside on the line of application of force, during its application (Figure 6).

### 3.3. Deformation of the Oral Mucosa (Strain)

The average values of deformation of the gum are shown in Table 2. For the results obtained it could be seen that the result was practically identical for the two models (right-angled and rounded) when simulated under the same stress of 0.2 N and 0.05 N, a result which is very similar to Von Mises values obtained for both the gum and the jaw bone.

Nevertheless, when the same morphology (right-angled and rounded) is compared for different stresses (0.05 N and 0.2 N) a difference of 0.2336 was observed, which as a percentage would be 25%.

### 3.4. Displacement of the Oral Mucosa

The average values of gum displacement may be found in Table 2. Here, displacement refers to the displacement of the gum by the application each of the simulated forces (0.05 N or 0.20 N) (Table 2).

On analysing the results, it was observed that there was a significantly greater result for the right-angled model with a stress applied to the suture of 0.2 N, that is, the highest stress applied. On the other hand, it was observed that the lowest value obtained was for the rounded angle model under 0.05 N of stress. A subsequent analysis of absolute values showed a difference of 24.3039 microns of displacement, which as a percentage implies that the model with right-angled blocks presents 81% more displacement than the right-angled model with an applied stress of 0.05 N.

Comparing the results in absolute values, we observe that the 0.2 N model with right-angled edges of the thin bone block differed 7.601 micron in displacement with respect to the thin bone block model with rounded angles when the same force is applied, which as a percentage is 25%.

If we compare the two models whose applied force is 0.05 N, a difference of 1.9506 micron of displacement may be observed between the two models, this difference being greater in the simulated right-angled thin bone block model, which percentage-wise is 25.55%, and which implies that the latter generates a greater displacement of the oral mucosa.

Similarly, if we compare the absolute values obtained of the right-angled thin bone block models for each of the two forces applied, we observe a difference of 22.3533 micron, which as a percentage is 74.53%, and making the same comparison for the rounded-angle models the difference is 16.6969 micron, which as a percentage is 74.59%. Therefore, in both cases the difference is practically identical when the force applied is increased.

## 4. Discussion

Dental implants are regarded as a reliable and safe treatment for restoring masticatory function and replacing lost teeth. The density and volume of bone that is accessible for the implant’s location, however, affects how well this treatment works [38,39,40,41].

There are no clinical studies that analyze the influence of block morphology on flap tension, and neither are there in vitro studies; the most similar studies are those by Jensen and Terheyden (2009) [7] and Stavola & Tunkel (2014) [19], however, none of them analyze the morphology of the block in post-surgical complications, which is why we propose a FEA, which is currently the most widely-used method to analyse stress, deformation and displacement when different graft designs and forces applied to the suture were combined during SBB procedure, in science and in industry. It has also been a proven technique for decades in resolving mechanical doubts associated with oral surgery and implantology. The use of the model applied in our study was prompted by a series of precautions with the aim of obtaining valid data. A 3D model was used that is in accordance with the majority of biomechanical studies in finite elements [42], and the geometric properties (with the exception of the morphology of the blocks, which was one of the independent variables) and mechanical ones were found to be identical for the four suppositions being tested [28,29].

Despite all this, our model is not without its limitations. In our case it was necessary to consider all the modelled materials, excepting the oral mucosa, as being homogenous and isotropic and having linear elasticity. However, authors such as O’Mahony et al. (2001) [43] and Bonnet et al. (2009) [44] claimed in their study that bone is heterogenous, anisotropic and shows no linear elasticity; although these properties are correct, the degree of complexity of the calculations would be far higher, and our simplification is in line with that of other studies published in literature [32,45,46]. Similarly, the oral mucosa was modelled as being isotropic and homogenous, but with visco-elastic characteristics according to the Prony Series [27], given that, as we know, the most internal oral mucosa shows resistance to deformation during the physiological load due to its being connected to the cortical bone [47]. However, in our study, the oral mucosa is not fixed, as it simulates recent surgery, although it is true that during the wound healing process it does become fixed again. In existing literature there is considerable controversy concerning the modelling properties of the oral mucosa. In Kim & Hong’s study (2016) [25] and Moldoveanu et al., (2020) [48], it was modelled as isotropic, homogeneous, as having linear elasticity and was given a Young’s Modulus of 2.8 MPa and a Poisson ratio of 0.4 and 19 MpA, 0,3 respectively. However, these same authors carried out a study very recently to predict changes after corrective jaw surgery in which, as was the case in our study, they modelled the oral mucosa as being visco-elastic [39]. Nevertheless, in existing literature studies may be found where elastic properties for soft tissue are applied which are very similar to those of Kim and his collaborators and those applied in our own test [38]. In the future, technological advances will make it possible to create a structural model made up of layers with different mechanical properties like vessels, collagen bundles and connective tissue to take into account the functions of each layer in order to accommodate visco-elasticity [27]. As regards the thickness of the gum, it was modelled at 1.5 mm, a value within the parameters considered by studies such as those by Kim & Hong. (2016) [25] and Moldoveanu et al. (2020) [49]; it is also in accordance with Wada et al. (2006) [50], who modelled the oral mucosa at 1 mm in the masticatory mucosa, which then goes progessively up to 3 mm in the mid-palatal area. However, the study of Lima et al. (2013) [48] modelled the oral mucosa at 1 mm, though their study dealt with the antero-inferior sector. Another of the limitations might be that of having ignored the inflammatory process suffered by any regenerative process, which causes an increase in the volume of soft tissue, reaching its peak 24–48 h following surgery. This process will also have effects on visco-elasticity, and these should be evaluated in future studies. Lastly, we used suture tensile strengths taken from existing literature, but they were simplified as we considered them to be static considering the resulting direction (45°) between the axis of the vestibular cortical and the occlusal area, which exerts stress during the wound healing process.

In general, despite these limitations, the results of our study coincide with what is to be expected, from both the physical point of view and the clinical: greater suture tension is associated with higher values of transferred stress to the bone and gum according to the Von Mises criterion, as well with as higher values for its deformation and displacement. Burjardt & Lang. (2009) [16] concluded that suture tension lower than 0.05 N would not cause dehiscence, whereas higher than 0.15 N would, although the thickness of the gum would also need to be considered. Similarly, Stavola & Tunkel (2014) [14], claimed that minimal stress should be applied to the edges of the flap, thereby confirming that any stress lower than 5 grammes (0.05 N) would not interfere with the wound healing process, a value that we ourselves have used as minimum applied force in our study. These clinical conclusions are in line with our mathematical findings, as we obtained 75% more Von Mises stress in the suppositions with 0.20 N stress applied to the incision (right-angled and rounded angled) when compared to the lowest suture tension applied (0.05 N). It is worth remembering that to achieve primary wound closure with no tension during the healing process it is necessary to prevent contamination and infection of the graft and, ultimately, protect the bone regeneration process [12,13,18].

Besides the tension applied to the suture, the other independent variable analysed in our study was the morphology of the block. Khoury & Hanser (2019) [10] claimed that using fine cortical bone blocks allows the atrophied ridge under reconstruction to adapt better to the morphology, and in turn favours the adaptation of the oral mucosa, when compared to the use of blocks of a conventional thickness (3 mm); similar results were obtained by Burjardt & Lang. (2009) [16] and Jensen & Terheyden, (2009) [7], as they confirmed that dehiscence does not only depend on the tension exerted by the suture on the closure of the mucosa, but also mentioned the morphology of the block itself. Our results confirmed this conclusion regarding the influence of the morphology, as with right-angled blocks we observed greater displacement of the oral mucosa; this effect may be due to the fact that that the right-angled blocks protrude more from the bone framework. We are not aware of any study published in existing literature which evaluates the biomechanical influence of the morphology of the block on a regenerative process. We chose a model of a 1 mm block due to the fact that as Khoury & Janser (2019) [10] y Restroy-Lozano et al. (2015) [51] affirmed, the bone block should be as thin as possible in order to facilitate vascularization and nutrition as well as to reinforce its mechanical stability. Although in other studies like that by Misch (2000) [5] blocks 4 mm thick were used, they failed to produce satisfactory results and even coated the block with membrane. As regards the arrangement of the blocks, we refer again to the study by Khoury & Hanser (2019) [10], which arranged them in an “L” shape, creating a casing, so as to prevent a resorption of the regeneration, as well as a possible encroachment of soft tissue (Khoury & Hanser, 2019) [10]. We performed two morphologies with totally contrasting block angles (right angle and rounded angle) and it is noteworthy that despite finding higher values of gum displacement for the right angle, we did not observe any significant difference in the Von Mises stress on the alveolar bone and oral mucosa between the two suppositions, with the same force, given that the area with the assumed conditions of contact (osseointegrated) does not tend to show high levels of stress.

Furthermore, the study by Mir-Mari et al. (2017) [52] claimed that the stability of the graft is the primary requisite for achieving satisfactory results in the regeneration, and to this end they affirmed that in order to do so it was necessary to use microscrews in block bone grafts. In our study, the area of greatest compression of the gums was where the miscroscrews were placed, which is due to this being slightly outside the bone framework. It is significant that there is greater tension on the most apical screw in the vestibular area, which coincides with the area of greatest tension on the stress distribution map. It is necessary to bear this factor in mind, as complications of the regeneration can also be due to a fenestration of the soft tissue at this level [53]. Looking at the analysis of the stress distribution map, the bone block with right angles showed a greater amount of stress at both a crestal and apical vestibular level in the area in contact with the microscrew [18]. In our study, the bone block was fixed with microscrews which were modelled with surgical steel, as affirmed by the studies [11,12,54]. Surgical steel is a less elastic material (193GpA) than titanium (114GpA) and therefore reduces the risk of a fracture in such a fine bone block (1 mm y 15GpA) in comparison with the use of titanium, which is routinely used for dealing with conventional 3 mm blocks. Regarding the screw connection, this was simplified, and a 100% passive fit with an intimate contact was assumed. The arrangement of the microscrews is important, as depending on where they are placed, they can generate an increase in tension. This supposition opens the door to further research in the future.

In short, the results of our study point to a greater tensile force applied as leading to an increase in stress, deformation and displacement being transferred to the gum, with the displacement also being influenced by the morphology of the block. Both suppositions should be taken into account at a clinical level, firstly in the correct manipulation of the volume and edges of the blocks, but also by means of a correct manipulation of the soft tissue. In this respect, PRI (Periosteal releasing incision) can play a crucial role in the immobility of the flap, and is an incision made in the vestibular flaps, both maxillary and mandibular [12,15,18]. Apart from the aforementioned, the study by De Stavola et al. (2021) [55], brought us up to date in dealing with the location of the incision: it informed us that we should make an incision in the vestibular area of as many millimetres as there are of defect that we are going to regenerate, and also that 4 mm should be added in order to achieve a correct closure of the wound. These 4 mm represent the height at which the mattress stitches will be made, as we will perform a double-plane suture, above which we will put simple stitches.

We can state that our results and the model generated prove useful and efficient; they open the door to possible future research, analyzing other independent variables, for instance the thickness of the bone block or the volume of the graft, different characteristics of the gum by biotopes or the number and arrangement of the osteosynthetic microscrews.

## 5. Conclusions

A greater suture force is related to an increase of tension on the regeneration, as well as producing in the oral mucosa an increase in tension, displacement and deformation.

There is a relationship between the morphology of the block and the appearance of dehiscence because of right-angled blocks. We observed greater displacement of the oral mucosa due to the fact that the right-angled blocks protrude more from the bone framework.

Therefore, analyzing the correct tension that the mucosa can bear and choosing a correct morphology of the block can lead us to obtain satisfactory results, preventing dehiscence.

## Figures and Tables

**Figure 1 materials-15-09039-f001:**
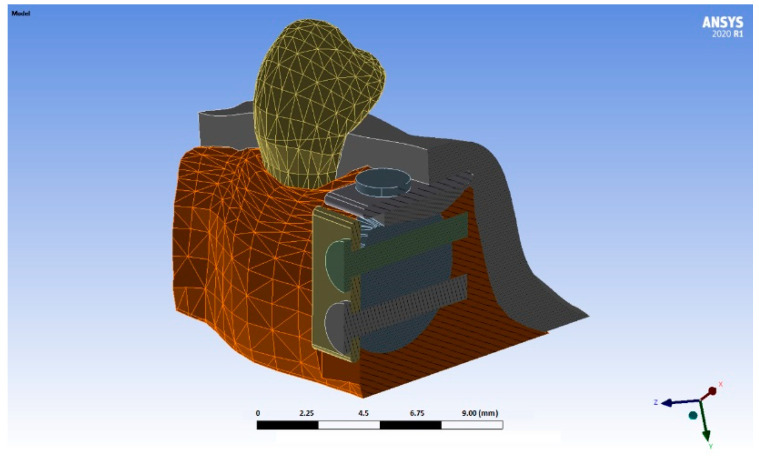
Simulation of a thin bone block with rounded edges (different colors are used to distinguish different parts of the model).

**Figure 2 materials-15-09039-f002:**
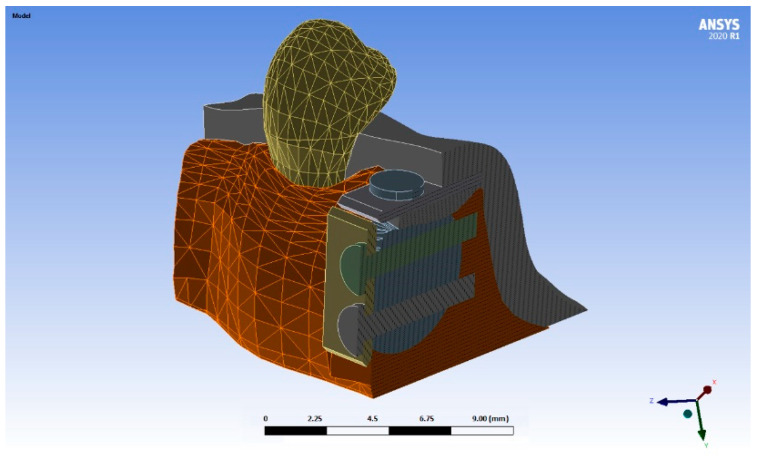
Simulation of a thin bone block with right-angled edges edges (different colors are used to distinguish different parts of the model).

**Figure 3 materials-15-09039-f003:**
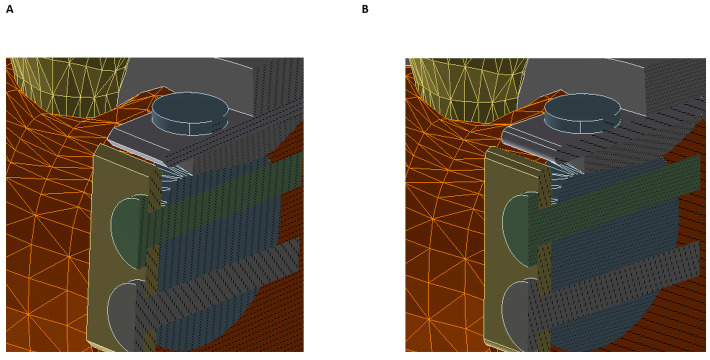
Zoom of: (**A**) Simulation of a thin bone block with right-angled edges, (**B**) Simulation of a thin bone block with rounded edges (different colors are used to distinguish different parts of the model).

**Figure 4 materials-15-09039-f004:**
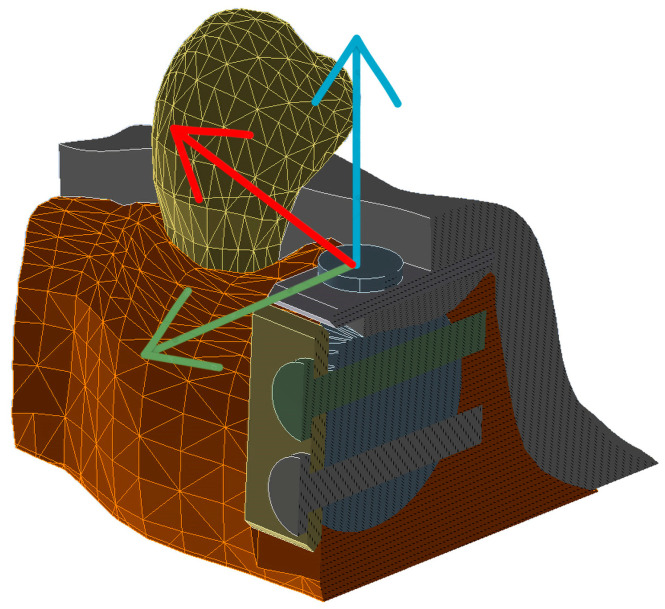
Tensile conditions applied to the model, red line is reffered to 45° according the direction is applied (green and blue lines indicate OX and OY axis respectively).

**Figure 5 materials-15-09039-f005:**
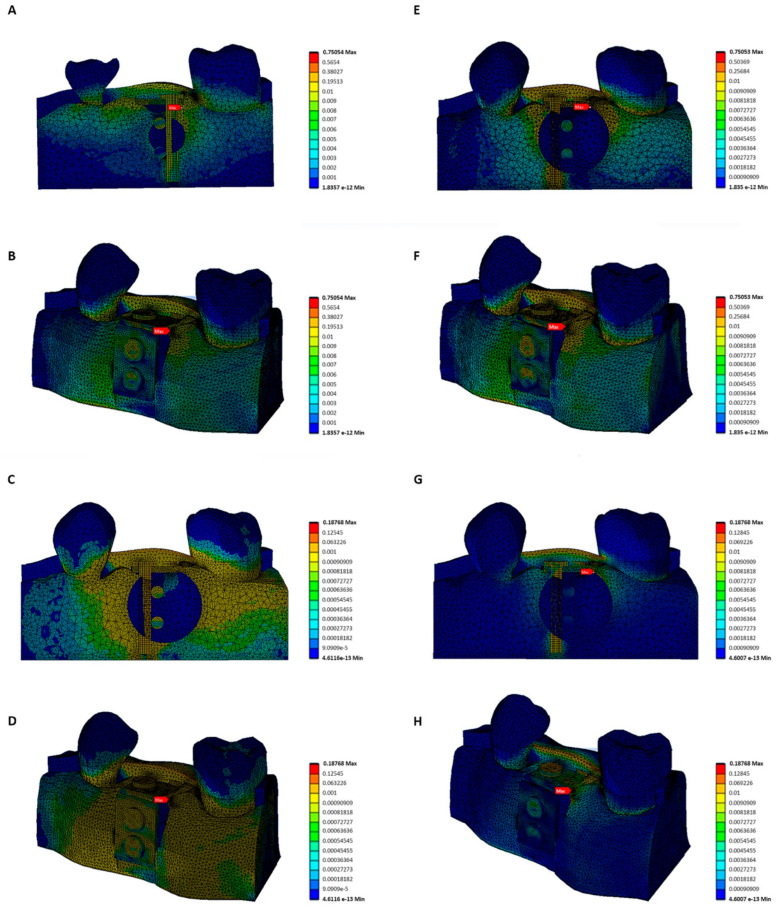
Stress distribution map for the different stresses and morphologies. (**A**,**B**) von Mises stress map for right angle 0.2 N; (**C**,**D**) von Mises stress map for right angle 0.05 N; (**E**,**F**) von Mises stress map for rounded angle 0.2 N; (**G**,**H**) von Mises stress map for rounded angle 0.05 N. The red arrow indicates the area of greatest transferred stress (MPa) according to the Von Mises criterion.

**Figure 6 materials-15-09039-f006:**
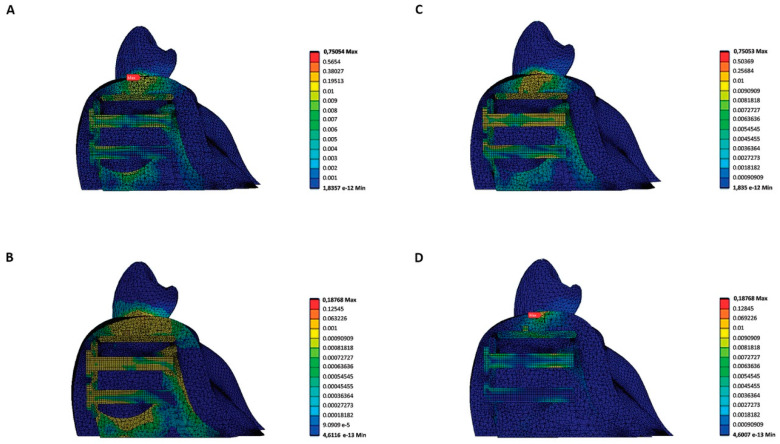
(**A**) lateral section of a stress distribution map for the different stresses and morphologies: A. stress distribution map for the right-angled model with 0.2 N force; (**B**) stress distribution map for the right-angled model with 0.05 N force; (**C**) stress distribution map for the rounded model with 0.2 force; (**D**) stress distribution map for the rounded model with 0.05 N force. The red arrow indicates the area of the greatest transferred stress (MPa) according to the Von Mises criterion.

**Table 1 materials-15-09039-t001:** Young Modulus and Poisson Ratios of each of the elements modelled.

MATERIAL	YOUNG MODULUS (GPa)	POISSON RATIO	REFERENCES
Cortical bone (Bone A2)	15	0.30	Geng et al., 2001 [28] Ma et al., 2014 [29]
Spongy bone (Bone A2)	1, 3	0.25	Geng et al., 2001 [28] Ma et al., 2014 [29]
Cortical blocks	15	0.30	Geng et al., 2001 [28]
Oral Mucosa	0.0028	0.35	Sawada et al., 2011 [27]
Microscrew (Surgical steel 316)	193	0.33	Ammar et al., 2011 [30]
Bone particles (Autologous bone A4)	0.231	0.25	Aguilar Henao et al., 2019 [31] Ma et al., 2014 [29]

**Table 2 materials-15-09039-t002:** Von Mises Stress (VM) on Jaw bone (Mpa), Von Mises Stress on Oral Mucosa (Mpa), Deformation (Micron) and displacement (Micron).

	0.2 C	0.2 R	0.05 C	0.05 R
**VM**	0.75054	0.75053	0.18768	0.18768
**VM Oral Mucosa**	0.75054	0.75053	0.18768	0.18768
**Deformation**	0.27117	0.27116	0.067812	0.067811
**Displacement**	29.991	22.384	7.6377	5.6871

## Data Availability

Not applicable.

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
