# Peer review of "A Biomechanical Analysis of the Influence of the Morfology of the Bone Blocks Grafts on the Transfer of Tension or Load to the Soft Tissue by Means of the Finite Elements Method"

_materials, 2022, doi:10.3390/ma15249039_

Round 1

Reviewer 1 Report

Model building is very important in the finite element method. Therefore, what is stated in result 2.5 should be explained by the method. Also, Tables 3-5 are not included in the text and cannot be peer-reviewed. Also, the cover letter is inserted after the bibliography and is not in the manuscript format specified. Therefore, we cannot recommend this paper for publication.

Reviewer 2 Report

Dear Authors,

your work is very interesting and well proposed and developed. I suggest to indicate in the title that the research work has been performed using finite element.

I advise you to add a section of the conclusions where you could summarize what emerges from your study and above all underline the clinical implications that emerge from the work.

Please reduce the length of discussion section: it's to long and redudant

Please consider to add these references to your:

Mechanical evaluation of the stability of one or two miniscrews under loading on synthetic bone. Journal of Functional Biomaterials2020, 11(4), 80

Reviewer 3 Report

In the SBB regeneration process, the applied suture tension and the morphology of the graft can be two independent variables, which affect the biomechanical behavior of the surgical wound. This biomechanical behavior may be closely related to a tendency to dehiscence and a failure to achieve primary wound closure. In this case, this manuscript tries to apply the finite elements method (FEM) analysis to explore the biomechanical behavior shown by the gum under the stress of regeneration with cortical bone blocks. It is an interesting report.

I am listing some concerns.

(1)     There are two levels of suture tension are applied (0.05N, 0.2N). In terms of tension level, what is the normal value in case of SBB regeneration?

(2)     Is it necessary to consider the affinity or interaction between graft block and wound substrate?

(3)     In terms of tensile conditions applied to the model, is it necessary to include the possible shear stress?

(4)     Is it possible to compare the simulation results with clinical outcome (by using either literature data or clinical data)?

(5)     Language issues: Page 2 Line 93, “…systems of [20]…”;

Round 2

Reviewer 1 Report

This manuscript is well written and the author's message is clear. This research contributes to this research field. I have some questions to fix.

Line129,130 : {Two suppositions were simulated, one with blocks with rounded edges (R), and the second with blocks with right-angled edges/Chamfer (C) (Figure 1)}

This article should be written like this:Two suppositions were simulated, one with blocks with rounded edges (R) (Figure 1)}and the second with blocks with right-angled edges/Chamfer (C) (Figure 2)}

Reviewer 3 Report

The authors have provided reasonable explanations to solve my previous concerns. And this revised version can be acceptable.
